# Pre-analytical errors during HIV viral load (HIV-VL) and early infant diagnosis (EID) testing in Cameroon

Marie Atsama-Amougou[1,2]*, Sabine Ndejo Atsinkou[3], Yagai Bouba[4],
Hamadou Amadou[3], Teclaire Elodie Ngo-Malabo[5,6], Nadine Nguendjoung Fainguem[4],
Julius Nwobegahay[7], Comfort Vuchas[8], Elong Elise[4], Bertrand Eyoum Bille[9],
David Donchi Nguiala[10], Emmanuel Biongolo[11], Boutgam Nadine Lamare[1],
Yakouba Liman[3], Marcel Tongo[1], Etienne Mpabuka[12], Ahidjo Ayouba[1],
Charles Kouanfack[1], Albert Franck Zeh Meka[3], Joseph Fokam[3,4]

**1** Research Center on Emerging and Re-emerging Diseases, Institute of Medical Research and Medicinal Plants Studies, Yaounde, Cameroon, **2** Laboratory of Pharmacology and Toxicology, University of Yaounde I, Yaounde, Cameroon, **3** Central Technical Group, National AIDS Control Committee (NACC), Yaoundé, Cameroon, **4** Chantal BIYA International Reference Centre for Research on HIV/AIDS Prevention and Management, Yaoundé, Cameroon, **5** Molecular Unit of Douala General Hospital, Douala, Cameroon, **6** Faculty of Sciences, University of Buea, Buea, Cameroon, **7** Centre for Research and Military Health (CRESAR), Yaoundé, Cameroon, **8** The Bamenda Center for Health Promotion and Research, Bamenda, Cameroon, **9** Retrovirology Laboratory, Laquintinie Hospital, Douala, Cameroon, **10** Dream Dschang Medical Center, Dschang, Cameroon, **11** Dream Nkolondom Medical Center, Yaounde, Cameroon, **12** ICAP Regional, Kigali, Rwanda

* marieamougou164@yahoo.com

## Abstract

### Background

HIV Viral Load (HIV-VL) and Early Infant Diagnosis (EID) play a pivotal role in the laboratory surveillance, monitoring of HIV/AIDS, and its elimination as a public concern. However, sample rejection due to sample nonconformity (SNC) resulting from inadequate collection, transportation, and management, especially during the pre-analytical phase, negatively affects laboratory performance. This study aimed to characterize errors observed during the pre-analytical phase of HIV-VL and EID testing across national reference laboratories in Cameroon and to identify factors associated with rejection.

### Methods

A descriptive and quantitative study of the nonconformities (NC) identified was collected from 11/01/2024 to 08/12/2024 in seventeen HIV reference laboratories, which constitute the national network of HIV-VL and EID testing coverage. For this study, the number of rejected samples, the reason for rejection, and the type of test ordered were recorded monthly.

**Data availability statement:** All relevant data are within the manuscript and its Supporting information files. The minimal data set underlying the findings of this study is provided as Supporting information file S1 Data, alongside S1–S2 Tables that summarize the main indicators analyzed.

**Funding:** The author(s) received no specific funding for this work.

**Competing interests:** All the authors declare that they have no competing interests.

## Results

During the study period, 326,885 and 38,354 specimens received for HIV viral load and EID. Of those 12,748 (3.9%) and 2.7% (1,039) were rejected. The SNC analysis indicates the presence of multiple errors or NC in some samples. For HIV viral load, our results indicate that specimen identification errors for viral load were the most common NC (63.14%; n = 8049; P = 0.031), followed by insufficient specimen volume (43.7%; n = 5571; P = 0.049) and quality errors, including hemolyzed specimens (27.8%; n = 3543; P = 0.054), and specimen transport packaging errors (9.1%; n = 1160; P = 0.069). For HIV EID, specimen rejections were primarily attributed to missing or mismatched identification on the request forms (37.12%, n = 386; P = 0.042), sample unavailability (13.4%; n = 139; P = 0.056), and information discrepancies (44.2%; n = 459; P = 0.033). The observed significant rejection rates for both HIV viral load and EID exceeded the established national rejection rate of <2% of errors. Our results suggest that corrective action is critical, along with the establishment of policies to detect and resolve preanalytical errors in Cameroon.

## Conclusion

Our findings highlight the high magnitude of preanalytical errors for HIV-VL and EID tests used in the testing and management of people living with HIV/AIDS in Cameroon. Therefore, the laboratory system should be strengthened to ensure high-quality patient services and support optimization. Suggestions for improvement include distributing a validated specimen-collection manual, creating electronic test-request forms, providing staff training, and regularly on-site supervising the use of available resources, all of which are necessary in this country.

## Introduction

Laboratory testing encompasses the pre-analytical, analytical, and post-analytical phases [1]. The pre-analytical phase, spanning from the initial test request to specimen preparation for analysis, is widely recognized as the most error-prone stage. Pre-analytical problems account for up to 70% of all laboratory errors [2,3]. This high error rate is largely driven by external factors and the manual handling steps, including test requests, specimen collection, labeling, storage, and transportation, which often occur outside of direct laboratory oversight [4,5]. If not adequately controlled, pre-analytical errors can compromise the reliability and accuracy of test results, delay clinical decision-making, increase healthcare costs, and ultimately undermine patient safety [6,7].

Testing for HIV Viral Load (HIV-VL) is the preferred method of monitoring the effectiveness of antiretroviral therapy (ART), while Early Infant Diagnosis (EID) enables HIV-infected infants to be identified and treated promptly. Together, these tests play a pivotal role in HIV surveillance, program monitoring, and epidemic control strategies [8,9].

In Cameroon, HIV remains a significant public health concern, with many individuals requiring lifelong ART and regular biological monitoring [10]. Over the past decade, sustained investment by the government and its partners has increased access to HIV diagnosis and treatment, improved ART coverage, and supported progress towards the global 95-95-95 targets [11]. HIV-VL and EID services are organized through a national network of 17 HIV reference laboratories, which provide testing coverage for all ten regions, receive specimens from a wide range of health facilities, and perform over 90% of all VL and EID tests for people living with HIV and HIV-exposed infants [12–14].

The complexity of this network, characterized by long referral distances, multiple collection sites, and several steps between sample collection and analysis, increases the risk of pre-analytical errors during specimen collection, labeling, storage, and transport [15–18]. Despite significant advancements in automation, laboratory information systems, and electronic reporting, which have enhanced the analytical and post-analytical phases of VL and EID testing, the pre-analytical stage remains a major source of error and poses substantial challenges to patient care [15–18].

Studies from other African settings have reported substantial VL and EID specimen rejection rates linked to pre-analytical problems, including mislabeling and identification errors, insufficient volume, hemolysis, inappropriate collection tubes, missing specimens, and incomplete request forms [19,20]. Such errors can result in repeated sampling, wasted resources, delayed ART initiation or modification, and missed opportunities for early infant treatment [19,20].

However, the scale, patterns, and causes of pre-analytical errors in HIV-VL and EID testing have not been systematically documented in Cameroon and many other Central African countries [21]. Although HIV reference laboratories routinely record specimen non-conformities, these data have not yet been analyzed at a national level in order to quantify rejection rates or identify where problems are concentrated along the referral system. This limits the national HIV program#39;s ability to design targeted, evidence-based interventions to improve sample quality, strengthen specimen transport, and optimize the use of limited laboratory resources [4,22].

The aim of this study was to investigate factors that contribute to pre-analytical errors in the HIV-VL and EID testing within Cameroon's network of HIV reference laboratories. Specifically, we sought to: (i) quantify the rates of VL and EID specimen rejection at national and regional levels; (ii) describe the main pre-analytical reasons for specimen rejection, and how these are distributed by region and health facility sample collection capacity; (iii) identify factors associated with specimen rejection in order to inform corrective actions for strengthening HIV laboratory systems, and support the effective follow-up of people living with HIV and HIV-exposed infants.

## Materials and methods

### Study design and setting

This cross-sectional descriptive study used routine quality-monitoring data from all 17 functional HIV reference laboratories in Cameroon between 11 January and 8 December 2024. These laboratories constitute the national HIV viral load (HIV-VL) and early infant diagnosis (EID) testing network. They receive plasma, whole blood, and dried blood spot specimens from HIV care and treatment facilities in all ten regions and perform the vast majority of HIV-VL and EID tests nationwide.

### Sampling and data collection

All 17 HIV reference laboratories performing HIV-VL and/or EID testing during the study period were included. All HIV-VL and EID specimens recorded as received in the participating reference laboratories between 11 January and 08 December 2024 were eligible for inclusion. Specimens for which no reason for rejection was documented were excluded from analyses of specific pre-analytical error patterns. For each laboratory, data on the number of specimens received and rejected, the type of test requested, the reason(s) for rejection, the region of origin, and the type of health facility were extracted from routine quality-monitoring logbooks using a structured data-abstraction form. Three complementary data

collection tools were used. First, a structured data-abstraction form was used by trained laboratory staff to extract aggregate information from routine quality-monitoring and problem-reporting logbooks in each reference laboratory, including the number of specimens received and rejected, the type of test requested (HIV-VL or EID), the documented reason(s) for rejection, the region of origin, and the type of referring health facility. Second, a brief self-administered questionnaire was completed once by the focal person at each reference laboratory to document site-level characteristics (test menus, annual testing volume, sample transport arrangements, and standard operating procedures for handling nonconforming specimens). Third, standardized national specimen collection and transport checklists accompanied individual shipments from HIV care and treatment centres to reference laboratories and were used by providers to verify proper sample collection, packaging, temperature control, and documentation before dispatch.

Specimens from peripheral HIV care and treatment centers were collected by trained nurses or laboratory technicians according to national guidelines and transported as outpatient samples to the reference laboratories, along with standardized test request forms and completed transport checklists. Upon receipt in the reference laboratories, all rejected specimens and the corresponding reason(s) for rejection were documented in problem-reporting logbooks by designated quality-assurance staff, together with the type of test ordered (HIV-VL or EID) and the specific pre-analytical discrepancies (hemolyzed specimen, insufficient volume, wrong specimen tube, mismatch between specimen and request form identifiers, inadequate temperature preservation, missing specimens, transport delay, or duplicate identification numbers assigned to different patients). All abstracted data from the 17 laboratories and questionnaires were entered into a centralized electronic database by the study coordination team for analysis.

### Reasons for sample rejection in reference laboratories

Samples were rejected according to pre-defined programmatic criteria. These included an inadequate sample volume, improper or incomplete patient labeling, or the use of a non-permanent marker. Samples that appeared compromised, such as those that were visibly hemolyzed or degraded, were also rejected. Other reasons included improper packaging, inappropriate transport temperature, insufficient drying of dried blood spot (DBS) cards prior to shipment for EID, and the absence of a completed patient or test request form. All reasons for rejection were recorded in the problem-reporting logbooks using standardized categories aligned with national guidelines.

### Program description

Cameroon's national HIV-VL and EID services are delivered via a network of 17 HIV reference laboratories. These laboratories receive plasma, whole blood, and dried blood spot specimens from HIV care and treatment facilities across the country#39;s ten regions. These laboratories are responsible for recording routine quality-monitoring indicators, including specimen non-conformities and reasons for sample rejection. Specimens are collected in accordance with national guidelines, packaged using standardized materials, and transported to the reference laboratories, along with harmonized request forms and transport checklists.

### Ethical consideration

The study protocol was submitted to the national ethics committee, which classified the work as a retrospective analysis of existing routine laboratory quality-monitoring data and determined that it did not constitute human-subjects research. The analysis focused exclusively on aggregate data on specimens received and rejected, reasons for rejection, and the origin of specimens (region and facility type). No personally identifiable patient data was collected.

### Statistical analysis

Statistical analysis was performed using IBM SPSS Statistics version 25.0 (IBM Corp., Armonk, NY, USA). Descriptive statistics were used to summarize the data, with continuous variables reported as means and standard deviations, and

categorical variables as frequencies and percentages. Overall and stratified specimen rejection rates were calculated for HIV-VL and EID by region of origin and by health facility sample-collection capacity. Stratified specimen rejection rates were calculated for HIV-VL and EID by region of origin and by health facility sample-collection capacity. Pearson's chi-square tests were used to compare proportions of rejected specimens across categories. Odds ratios with 95% confidence intervals were estimated using logistic regression to identify the factors associated with specimen rejection. A p-value of less than 0.05 was considered statistically significant. Specimens with missing information on the specific reason for rejection were excluded from the analyses of pre-analytical error patterns, but were included in the overall counts of specimens received and rejected.

## Results

### Distribution of samples attributed in 2024 for HIV viral load and EID testing

This cross-sectional study examined 326,885 specimens submitted for HIV-VL testing and 38,354 for EID across 17 HIV reference laboratories in Cameroon in 2024 (Table 1). Most of the specimens originated from high-volume health facilities that provide ART services to large numbers of people living with HIV. The Centre region accounted for the largest proportion of both HIV-VL (187,631; 57.4%) and EID (24,830; 64.7%) specimens. Overall, 12,748 (3.9%) HIV-VL specimens were rejected, with rejection rates ranging from 2.4% in the Far North region to 5.6% in the Centre region. Meanwhile, 1,039 (2.7%) of the 38,354 EID specimens were rejected, with rejection rates ranging from 2.2% to 2.9% (Table 1).

### Distribution of reasons for sample rejection during the pre-analytical stage in HIV viral load and EID testing laboratories

Of all specimens received, 209 HIV-VL samples and 59 EID dried blood spot (DBS) samples for which the reason for rejection was not documented were excluded from the analysis of specific pre-analytical error patterns. Of the specimens with a documented reason for rejection, 3.9% of HIV-VL samples (12,740 out of 326,676) were rejected, with rejection rates in different regions ranging from 2.4% to 5.6%. Meanwhile, 2.7% of EID samples (1,034 out of 38,295) were rejected, with rejection rates ranging from 2.2% to 3.3%. Some specimens had more than one recorded pre-analytical error. In multivariable analysis, the adjusted odds ratios for high-volume facilities remained close to the crude estimates, indicating that adjustment had only a limited impact on these associations

For HIV-VL, the most common reason for rejection was identification problems (mislabelling or duplicate identifiers) (63.14%; n = 8,049; p = 0.031), followed by insufficient sample volume (43.7%; n = 5,571; p = 0.049), and quality issues such as haemolysed samples (27.79%; n = 3,541) and packaging errors (9.06%; n = 1,160). For EID, the main reasons

**Table 1. Distribution of samples received and rejected per region attributed to HIV viral load and EID testing in 2024 in Cameroon.**

| Region | HIV-VL: Specimens received, n | HIV-VL: Specimens rejected, n (%) | EID: Specimens received, n | EID: Specimens rejected, n (%) |
|---|---|---|---|---|
| Center | 187,631 | 10,508 (5.6) | 24,830 | 734 (2.9) |
| Littoral | 11,834 | 367 (3.1) | 5,873 | 129 (2.2) |
| Far North | 30,100 | 722 (2.4) | – | – |
| North West | 33,000 | 1,056 (3.2) | – | – |
| South West | 32,220 | 805 (2.5) | 7,651 | 176 (2.3) |
| East, South, Adamawa, West † | 32,100 | 998 (3.1) | – | – |
| Total | **326,885** | **12,748 (3.9)** | **38,354** | **1,039 (2.7)** |

†Data for early infant diagnosis were not available for all regions. EID, early infant diagnosis; HIV-VL, HIV viral load.

for rejection were missing samples and/or request forms, followed by incomplete or inconsistent identification information (Table 2).

## Factors affecting pre-analytical errors

Of the 63.14% (8,049/12,748) of HIV-VL specimens that were rejected due to identification or labelling errors, 44.5% (3,581/8,049) could not be reliably identified because non-permanent markers were used, 21.7% (1,746/8,049) had duplicate identifiers, and 33.2% (2,672/8,049) had missing identifiers. Additionally, 4.0% (322/8,049) were listed on the checklist, but the physical specimens were unavailable.

When the data were stratified by test type, region of origin, and health facility sample-collection capacity, differences in rejection rates were observed between specimens collected in the same region as the testing laboratory and those referred from other regions, as well as between high- and low-volume facilities (Table 2). The proportion of pre-analytical errors was higher among specimens originating from other regions than among those collected within the testing region (76.4% versus 23.6% for HIV-1 VL and 69.1% versus 30.9% for EID). Errors were also more frequent among specimens from high-volume facilities than from low-volume facilities (55.6% versus 44.4% for VL and 50.7% versus 49.3% for EID). Overall, the proportion of rejected specimens ranged from 2% to 4% for HIV-VL and from 1% to 3% for EID. The highest rejection rates were observed among specimens collected outside the testing region and in facilities with a high sample-collection capacity (Table 2).

## Stratification of data by the type of requested analysis and the region or healthcare capacity of the sample collection

Among the 63.14% (8,049/12,748) of HIV-VL specimens rejected due to identification or labelling errors, 44.5% (3,581/8,049) could not be reliably identified because non-permanent markers were used, 21.7% (1,746/8,049) carried duplicate identifiers, 33.2% (2,672/8,049) lacked identifiers, and 4.0% (322/8,049) were listed on the checklist but the physical specimens were not available (Table 2).

When the data were stratified by test type, region of origin, and health facility sample collection capacity, significant differences in rejection rates were observed (Table 2). The proportion of pre-analytical errors was higher among specimens originating from outside the testing region than among those collected within the same region (76.4% versus 23.6%

Table 2. Factors associated with specimen rejection for HIV-VL and EID testing, by region of origin and facility sample-collection capacity.

| Test type | Factor category | Category | Specimens received, n (%) | Specimens rejected, n (%) | Crude OR (95% CI) | Adjusted OR † (95% CI) | p-value |
|---|---|---|---|---|---|---|---|
| HIV-VL | Region of specimen origin | Same region as testing laboratory | 257,324 (78.8) | 6,690 (2.6) | Reference | Reference | — |
| HIV-VL | Region of specimen origin | Other regions | 69,561 (21.3) | 3,756 (5.4) | 2.21 (1.92–5.87) | 2.5 (2.1–3.0) | 0.0009 |
| HIV-VL | Facility sample-collection capacity | Low-volume facilities | 52,628 (16.1) | 1,210 (2.3) | Reference | Reference | — |
| HIV-VL | Facility sample-collection capacity | High-volume facilities | 274,257 (83.9) | 13,987 (5.1) | 4.17 (2.67–9.57) | 4.2 (2.7–9.6) | 0.0006 |
| EID | Region of specimen origin | Same region as testing laboratory | 789 (76.2) | 19 (2.4) | Reference | Reference | — |
| EID | Region of specimen origin | Other regions | 246 (23.8) | 8 (3.3) | 4.24 (2.58–11.12) | 2.1 (1.0–4.5) | 0.052 |
| EID | Facility sample-collection capacity | Low-volume facilities | 175 (16.9) | 4 (2.3) | Reference | Reference | — |
| EID | Facility sample-collection capacity | High-volume facilities | 862 (83.4) | 30 (3.5) | 5.12 (3.72–10.31) | 5.1 (3.7–10.3) | 0.0002 |

† Adjusted for region of specimen origin and facility sample-collection capacity (logistic regression models). OR, odds ratio; CI, confidence interval; EID, early infant diagnosis; HIV-VL, HIV viral load.

for HIV-VL; p = 0.022, and 69.1% versus 30.9% for EID; p = 0.033). Errors were also more frequent in specimens from high-volume facilities than from low-volume facilities (55.6% versus 44.4% for HIV-VL; p = 0.042, and 50.7% versus 49.3% for EID; p = 0.052). Overall, the proportion of rejected specimens ranged from 2% to 4% for HIV-VL and from 1% to 3% for EID, with the highest rejection rates observed for specimens collected outside the testing region and in high-volume facilities. In these facilities, specimens were approximately five times more likely to be rejected than in facilities serving fewer patients (Table 2).

## Discussion

This cross-sectional study, conducted across the national network of HIV reference laboratories in Cameroon, identified the main reasons for specimen rejection and quantified pre-analytical errors in HIV viral load (HIV-1 VL) and early infant diagnosis (EID) testing. Although national guidelines recommend a maximum specimen rejection rate of less than 2% per month, we observed rejection rates of 3.9% for HIV-VL and 2.7% for EID, indicating that pre-analytical quality targets are not being met [23]. These findings underscore the need for targeted interventions to strengthen sample collection, labelling, packaging, and transport processes in health facilities responsible for the biological monitoring of people living with HIV in Cameroon.

Despite the advent of automation in HIV-VL and EID laboratories and sustained efforts to improve HIV molecular testing, analytical quality errors continue to occur throughout the laboratory testing cycle [1,2,24]. It is well known that improving laboratory quality requires a proactive approach to identifying, documenting, and tracking errors that can undermine test result reliability [4–6].

The rejection rates observed in our study are relatively high for HIV-VL and comparable to those reported in Ethiopia (3.6%) and in other settings that have documented a high prevalence of pre-analytical errors [4,5,25]. For EID, our overall rejection rate falls within the range reported in Zimbabwe (2–7% depending on the region), but still exceeds national targets [5]. Differences between countries may reflect variation in health system structure, the maturity of sample transport networks, and the extent of decentralisation of HIV services, as well as differences in training, supervision, and the implementation of quality assurance systems. In some settings, stronger sample referral systems and more robust supervision may help to keep rejection rates closer to programme benchmarks, whereas in others, persistent logistical and staffing constraints may contribute to higher levels of nonconformity. Where such contextual factors are not well documented, they remain plausible but unconfirmed explanations that warrant further investigation.

The current study found that the most common pre-analytical non-conformities or reasons for rejection in HIV-VL and EID testing were sample identification errors or mismatches, insufficient volume, and haemolysed samples. Our findings are consistent with studies from Ethiopia, Zimbabwe, and other settings, which have also highlighted identification errors, low sample volume, and sample quality issues as leading causes of rejection [4,5,26–28]. By contrast, one Ethiopian study reported that the predominant reason for specimen rejection was the use of inappropriate sample collection containers [4], suggesting that specific error profiles may vary according to local practices, available supplies, and adherence to standard operating procedures.

We also found that distance from the testing laboratory and high health-facility sample volume were important factors associated with specimen rejection. Specimens collected in regions different from the location of the HIV-VL or EID testing laboratory were more likely to be rejected than those collected within the same region. This association may reflect longer transport times, increased risk of temperature excursions, and more complex referral pathways involving multiple handling points, all of which can increase the likelihood of labeling errors, delays, or physical damage to samples. Our results are similar to findings from Nigeria and Zimbabwe, where samples referred from distant or peripheral facilities had higher rejection rates [5,29].

Remarkably, specimens collected in high-volume healthcare facilities, where most patients receive HIV care, were several times more likely to be rejected than those from lower-volume facilities. This pattern may be explained by

higher staff workload, time pressure, frequent staff rotations, and possible gaps in ongoing training or supervision in busy facilities, which together can compromise adherence to specimen collection and labelling protocols. Our findings align with Nigerian data, indicating that the majority of rejected samples originated from facilities that serve large patient populations [29]. These observations suggest that quality-improvement efforts should prioritise high-volume and remote facilities, for example, through targeted mentoring, task-specific training, simplified job aids, and strengthened sample transport systems.

The high incidence of specimen rejection in this study highlights the need for preventive strategies to minimise the risks associated with incorrect or inadequate specimen collection, including delayed results, diagnostic errors, adverse medication events, and compromised patient safety. Addressing pre-analytical errors is particularly critical in the context of HIV programmes, where timely and accurate viral load and EID results are essential for monitoring treatment effectiveness, guiding regimen changes, and ensuring early ART initiation in infants.

## Limitations of the study

Interview-based information depended on responses from laboratory personnel, and response bias is possible because data were collected during routine work. However, observational data helped to minimise this bias. In addition, this study has important limitations related to the source and structure of the programmatic data used. All variables were abstracted solely from the specimen reception registers of HIV-VL and EID laboratories, which are designed for routine sample tracking rather than comprehensive clinical or socio-demographic documentation. Consequently, key patient-level characteristics (including detailed age, sex for all specimens, education, occupation, and socioeconomic status) were not systematically recorded and could not be incorporated into the analysis. This limitation prevented a more granular assessment of whether specimen rejection patterns differ across population subgroups and may mask inequities in access to high-quality viral load and EID services among the most vulnerable patients. Regarding both HIV-VL and EID, there were no available data to track the interval between sample collection and informing the mother or patient that the sample had been rejected. Such data could have indicated whether sample rejection adversely affects the timeliness of ART initiation in infants or the need for regimen adjustments in adults.

## Conclusion

Despite substantial progress in strengthening the analytical phase of HIV-VL and EID testing, our findings show that pre-analytical errors remain frequent in Cameroon and result in specimen rejection rates that exceed national quality targets. These errors are largely due to identification problems, insufficient sample volume, and poor sample quality, particularly in high-volume facilities and for referred specimens.

To address these gaps, the HIV laboratory system needs to be reinforced with interventions that directly target the observed sources of error. First, implementing and disseminating a validated national sampling manual and related job aids would standardise procedures for specimen collection, labelling, packaging, and transport, thereby reducing variability and poor adherence to existing guidance. Second, introducing electronic test request forms integrated with laboratory information systems would help prevent mislabelling and "other information discrepancies" by improving the completeness, legibility, and consistency of patient and test information. Third, regular competency-based training and refresher courses for staff involved in specimen collection and handling especially in high-volume and remote facilities—would address errors linked to inadequate skills, high workload, and staff turnover. Finally, routine on-site supervision and mentoring focused on pre-analytical quality indicators, including specimen rejection rates, would support continuous quality improvement by identifying recurrent problems and ensuring that corrective actions are implemented and sustained. Together, these measures can reduce pre-analytical errors, improve laboratory performance in terms of specimen acceptance and quality, and ultimately strengthen HIV biological monitoring in Cameroon.

## Supporting information

**S1 Table. Regional Distribution of sample received and rejected.**
(XLSX)

**S2 Table. Rejection Reasons for VL.**
(XLSX)

**S3 Table. Rejection Reasons for EID.**
(XLSX)

**S4 Table. Logistic Regression.**
(XLSX)

## Acknowledgments

We would like to thank all the staff of the HIV reference laboratories for their involvement in quality control records in this study.

## Author contributions

**Conceptualization:** Marie Atsama-Amougou, Boutgam Nadine Lamare.

**Data curation:** Emmanuel Biongolo, Marcel Tongo, Etienne Mpabuka.

**Formal analysis:** Marie Atsama-Amougou, Sabine Ndejo Atsinkou, Hamadou Amadou, Elong Elise, Emmanuel Biongolo, Etienne Mpabuka, Charles Kouanfack, Albert Franck Zeh Meka.

**Investigation:** Hamadou Amadou, Charles Kouanfack.

**Methodology:** Marie Atsama-Amougou, Yagai Bouba, Teclaire Elodie Ngo-Malabo, Nadine Nguendjoung Fainguem, Comfort Vuchas, Bertrand Eyoum Bille, David Donchi Nguiala, Boutgam Nadine Lamare, Etienne Mpabuka, Ahidjo Ayouba, Albert Franck Zeh Meka, Joseph Fokam.

**Project administration:** Sabine Ndejo Atsinkou, Hamadou Amadou, Etienne Mpabuka.

**Software:** Yagai Bouba, Bertrand Eyoum Bille, Ahidjo Ayouba, Joseph Fokam.

**Supervision:** Julius Nwobegahay.

**Validation:** Marie Atsama-Amougou, Elong Elise, Yakouba Liman, Ahidjo Ayouba.

**Writing – original draft:** Marie Atsama-Amougou, Etienne Mpabuka.

**Writing – review & editing:** Marie Atsama-Amougou, Sabine Ndejo Atsinkou, Yagai Bouba, Hamadou Amadou, Teclaire Elodie Ngo-Malabo, Nadine Nguendjoung Fainguem, Julius Nwobegahay, Comfort Vuchas, Elong Elise, Bertrand Eyoum Bille, David Donchi Nguiala, Emmanuel Biongolo, Boutgam Nadine Lamare, Yakouba Liman, Marcel Tongo, Etienne Mpabuka, Ahidjo Ayouba, Charles Kouanfack, Albert Franck Zeh Meka, Joseph Fokam.

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
