## [Decision Letter · Decision Letter 0]

12 Nov 2025

Dear Dr. AMOUGOU-ATSAMA,

Thank you for submitting your manuscript to PLOS ONE. After careful consideration, we feel that it has merit but does not fully meet PLOS ONE’s publication criteria as it currently stands. Therefore, we invite you to submit a revised version of the manuscript that addresses the points raised during the review process.

We look forward to receiving your revised manuscript.

Kind regards,

Abiola Olaleye, MBBS, MPH, FWACP

Academic Editor

PLOS ONE

2. We note that Figure 1 in your submission contain [map/satellite] images which may be copyrighted. All PLOS content is published under the Creative Commons Attribution License (CC BY 4.0), which means that the manuscript, images, and Supporting Information files will be freely available online, and any third party is permitted to access, download, copy, distribute, and use these materials in any way, even commercially, with proper attribution. For these reasons, we cannot publish previously copyrighted maps or satellite images created using proprietary data, such as Google software (Google Maps, Street View, and Earth). For more information, see our copyright guidelines: http://journals.plos.org/plosone/s/licenses-and-copyright.

Reviewers' comments:

Reviewer's Responses to Questions

**Comments to the Author**

1. Is the manuscript technically sound, and do the data support the conclusions?

Reviewer #1: Partly

Reviewer #2: Partly

Reviewer #3: Yes

2. Has the statistical analysis been performed appropriately and rigorously?

Reviewer #1: No

Reviewer #2: No

Reviewer #3: I Don't Know

3. Have the authors made all data underlying the findings in their manuscript fully available?

Reviewer #1: No

Reviewer #2: No

Reviewer #3: Yes

4. Is the manuscript presented in an intelligible fashion and written in standard English?

Reviewer #1: No

Reviewer #2: Yes

Reviewer #3: No

Reviewer #1: 1. While a useful manuscript, I felt there was room for improved clarity around the methodology used, and the analysis done in order to share best practices of analysing this important issue with other countries

2. the authors could include some additional variables in inferential analysis beyond the samples collected within or outside the region of the lab - details are in the attached review.

3. have not seen the questionnaire to or a description of variables collected -

4. Overall he manuscript would need some revisions to have direct and articulate sentences to ensure clarity for the reader- some statements included assume the reader knows what the author referes to e.g the phrase "non- permanet marker ID)

Reviewer #2: Identification of common pre-analytical errors during the HIV viral load and EID testing in Cameroon: A call for strengthening the laboratory systems to support follow-up people living with HIV/AIDS.

Review Comments – Dr. Nanyeenya Nicholus

A. General Comments

This is a very critical study which seeks to provide more knowledge about pre-analytical errors in Viral load (VL) and EID testing in Cameroon in West Africa. Both VL and EID in West Africa are being scaled up, and facing different challenges like increased sample rejection due to pre-analytical errors. Hence such a study will produce key information that might inform key policy decisions in the entire region. However the study still need significant improvements in all the sections. The introduction needs to be revised to set the scene for the study. The methods are grossly under-reported and need to be overhauled with a comprehensive data analysis, with support from a good Biostatistician to bring out the study results well. Consequently, the results section also needs an overhaul, re-writing the section to address the different comments below.

B. Specific Comments

1. Introduction: The authors have written a fairly good and concise introduction. However there are a number of issues that need to be rectified, to make the introduction better and these include;

a) The introduction does not set the scene for the study. Since the study is about pre-analytical errors for HIV VL and EID, there is need to set up the scene, starting with a brief description of the i) burden of HIV in Cameroon, ii) progress made in treatment coverage, iii) progress made in laboratory with a focus on diagnosis in infants (EID) and monitoring of ART (VL), hence this introduces the concepts of VL and EID, iv) the challenges faced, including pre-analytical errors, and this can then be described comprehensively, and then v) how the study is important. The study introduction does not chronologically flow well, and misses linkage of all these aspects. Revise the introduction to include all these aspects.

b) The literature citation in the introduction is also sub-optimal. There is need to do a thorough literature review around the pre-analytical errors in VL and EID testing and use them to build a case here.

c) A brief description of the VL and EID testing services and their coverage would also strengthen the introduction.

d) This is an important topic, but the authors do not elaborate well about the problem in this study, and actually the actual problem being addressed remains slightly unclear.

i) Hence there is need to clearly define the problem in this study, and its magnitude

ii) What has been done before about the problem and the risk of not addressing the problem? This should all be clear in the introduction.

2. Methods: The methods section is generally very sub-optimal, with several issues to address;

a) The sampling and data collection section is very unclear. How was the sampling done in each facility? What was the sample size? Were all the eligible samples assessed? What was the inclusion and exclusion criteria for the samples? How was the semi-administered questionnaires developed, piloted and used? Which key variables were collected?

b) The section of program description (how the programs run) could be removed from the section of sampling and data collection and made to stand alone. This will make the methods section better.

c) The statistical analysis section is also missing the key analysis techniques that were used, which makes it very hard to understand how the data was analysed. Was there missing data? And if yes, how was it managed?

d) For the ethical consideration, I am not sure whether it was right not to review the study simply because it was retrospective. I would rather think that the study would have been reviewed by the Ethics Committee but instead the informed consent would have been waived given its retrospective nature. You will kindly need to clarify this, as it might raise ethical concerns.

3. Results: In the results section, the descriptive statistics are sub-optimally presented. Including a table with key variables like age and gender is key for the results section.

4. The data analyses also miss detailed analysis to understand how key variables like age and gender among others affect sample rejection.

5. Both tables 1 and 2 are not clearly drawn and they are very difficult to follow-through and understand. For table 1, normally the frequency/number is together with the percentage in brackets for example n (%) i.e 10508 (5.6). Kindly re-draw these tables so that they are easy to understand.

6. I generally feel the authors need to work with a good Biostatistician and re-analyze the results well using good statistical techniques, and also re-write the entire results section to improve it.

Reviewer #3: The authors didn't provide a considerable depth of the statistics used in the article under the section on the statistical analysis.

Line 201-205: What was the comparative group for each of the comparisons, for example, line 201 to 2016? The n for mislabelling and identification line 202 should be edited to “12,748” rather than “12 748”.

Line 228: Why do you need to reference table 2 here when that analysis isn't in table 2? If you're referring to the one in the supplementary material, it should be clearly labelled.

Line 278: The Authors mention that your study findings were different from those of a study in Ethiopia, but do not elaborate on why you think you have differing results.

Line 290: The use of the word interviewed means something different from what I think they were meant to say.

Line 229-243: It is not entirely clear to me whether these results are derived from Table 3, as this connection is not explicitly stated.

In Table 3, what was the reference group for each of those Odds Ratios?

There is a need for grammatical improvements.

.

Reviewer #1: No

Reviewer #2: **Yes:**Nanyeenya NicholusNanyeenya NicholusNanyeenya NicholusNanyeenya Nicholus

Reviewer #3: **Yes:**Joseph B. SempaJoseph B. SempaJoseph B. SempaJoseph B. Sempa

---

## [Author Response · Author response to Decision Letter 1]

8 Jan 2026

Identification of common pre-analytical errors during the HIV viral load and EID testing in Cameroon: A call for strengthening the laboratory systems to support follow-up people living with HIV/AIDS.

Review Comments – Dr. Nanyeenya Nicholus

A. General Comments

This is a very critical study which seeks to provide more knowledge about pre-analytical errors in Viral load (VL) and EID testing in Cameroon in West Africa. Both VL and EID in West Africa are being scaled up, and facing different challenges like increased sample rejection due to pre-analytical errors. Hence such a study will produce key information that might inform key policy decisions in the entire region. However the study still need significant improvements in all the sections. The introduction needs to be revised to set the scene for the study. The methods are grossly under-reported and need to be overhauled with a comprehensive data analysis, with support from a good Biostatistician to bring out the study results well. Consequently, the results section also needs an overhaul, re-writing the section to address the different comments below.

B. Specific Comments

1. Introduction: The authors have written a fairly good and concise introduction. However, there are a number of issues that need to be rectified, to make the introduction better and these include;

a) The introduction does not set the scene for the study. Since the study is about preanalytical errors for HIV VL and EID, there is need to set up the scene, starting with a brief description of the i) burden of HIV in Cameroon, ii) progress made in treatment coverage, iii) progress made in laboratory with a focus on diagnosis in infants (EID) and monitoring of ART (VL), hence this introduces the concepts of VL and EID, iv) the challenges faced, including pre-analytical errors, and this can then be described comprehensively, and then v) how the study is important. The study introduction does not chronologically flow well, and misses linkage of all these aspects. Revise the introduction to include all these aspects.

We thank the reviewer for this constructive feedback. We have substantially revised the introduction, providing a stronger contextual background and a clearer, more logical flow that situates the study firmly within Cameroon's national HIV response framework.

(i) Burden of HIV in Cameroon:

We have added a concise overview of the epidemiological context of HIV in Cameroon, including prevalence trends and the population groups most affected (Lines 91 to 95).

In Cameroon, HIV remains a significant public health concern, affecting a large proportion of the population who require lifelong ART and regular biological monitoring (10). Over the past decade, sustained investment from the government and its partners has increased access to HIV diagnosis and treatment, improved ART coverage, and advanced progress towards the global 95-95-95 targets.

(ii) Progress in treatment coverage:

We have introduced a paragraph that highlights the country’s progress towards the 95-95-95 targets, linking improved ART coverage to national and international programme investments (Lines 92 to 95).

Over the past decade, sustained investment from the government and its partners has increased access to HIV diagnosis and treatment, improved ART coverage, and advanced progress towards the global 95-95-95 targets.

(iii) Progress in laboratory systems (VL and EID):

We elaborated on the scale and structure of Cameroon’s HIV reference laboratory network, emphasising its critical role in supporting early infant diagnosis (EID) and viral load (VL) testing for monitoring antiretroviral therapy (ART) outcomes (Lines 96 to 100).

In this country, VL and EID services are organized through a national network of 17 HIV reference laboratories, which provide testing coverage for all ten regions. These laboratories receive specimens from a wide range of health facilities and perform over 90% of all VL and EID tests for people living with HIV on antiretroviral therapy (ART) and HIV-exposed infants. This makes them critical for achieving and maintaining viral suppression at a population level.

(iv) Challenges and pre-analytical errors:

We provided a seamless transition to describe the systemic challenges, particularly in the pre-analytical phase, which remains the most error-prone due to logistical and operational constraints in specimen collection and transportation (Lines 91-95; 99-100, 101-107).

“In Cameroon, HIV remains a significant public health concern, affecting a large proportion

of the population who require lifelong ART and regular biological monitoring (10). Over the past decade, sustained investment from the government and its partners has increased access to HIV diagnosis and treatment, improved ART coverage, and advanced progress towards the global 95-95-95 targets (11). In this country, VL and EID services are organized through a national network of 17 HIV reference laboratories, which provide testing coverage for all ten regions. These laboratories receive specimens from a wide range of health facilities and perform over 90% of all VL and EID tests for people living with HIV on antiretroviral therapy (ART) and HIV-exposed infants (12). This makes them critical for achieving and maintaining viral suppression at a population level (13,14).

The complexity of this network, involving long referral distances, multiple collection sites, and several steps between sample collection and analysis (15–17), also increases the risk of pre-analytical errors during specimen collection, labelling, storage, and transport (18). Despite important progress in automation and laboratory information systems, as well as electronic reporting, which have strengthened the analytical and post-analytical phases of VL and EID testing, the pre-analytical stage remains a major source of error and continues to pose significant challenges(15–17) and detrimental consequences for patient care (18).

(v) Study rationale and importance:

We strengthened the concluding paragraph to clearly justify the importance of the study, emphasising that a national-level analysis of pre-analytical errors had not yet been conducted and that such findings are essential for guiding quality improvement interventions (Lines 114 to 121) and lines 122 to 127.

“However, in Cameroon and many other Central African countries, the scale, patterns, and causes of pre-analytical errors in HIV-VL and EID laboratories testing have not yet been systematically documented (22). Although HIV reference laboratories routinely record specimen non-conformities, these data have not yet been analyzed at a national level to quantify rejection rates or identify where problems are concentrated in the referral system. This limits the national HIV program's ability to design targeted, evidence-based interventions to improve sample quality, strengthen the sample transport system, and optimize the use of limited laboratory resources (4,23)”.

“The aim of this study was therefore to investigate factors contributing to pre-analytical errors in the HIV-VL and EID testing within Cameroon’s HIV reference laboratories network. Specifically, we sought to: (i) quantify the rates of VL and EID specimen rejection at national and regional levels; (ii) describe the main pre-analytical reasons for specimen rejection and how these are distributed by region and health facility collection capacity; (iii) identify factors associated with specimen rejection to inform corrective actions for strengthening HIV laboratory systems and supporting the effective follow-up of people living with HIV and HIV-exposed infants”.

b) The literature citation in the introduction is also sub-optimal. There is a need to do a thorough literature review around the pre-analytical errors in VL and EID testing, and use them to build a case here.

Regarding the comment on the sub-optimal literature citation in the introduction, we thank the reviewer for this important observation. In the revised version of the Introduction, we have strengthened the literature review on pre-analytical errors in HIV viral load and early infant diagnosis testing and used these data to better build the case for our study.

-First, we expanded the opening paragraphs to more clearly summarise published evidence that the pre-analytical phase is the most error-prone stage of the laboratory testing process (Lines 114 to 121),

“However, in Cameroon and many other Central African countries, the scale, patterns, and causes of pre-analytical errors in HIV-VL and EID laboratories testing have not yet been systematically documented (22). Although HIV reference laboratories routinely record specimen non-conformities, these data have not yet been analyzed at a national level to quantify rejection rates or identify where problems are concentrated in the referral system. This limits the national HIV program's ability to design targeted, evidence-based interventions to improve sample quality, strengthen the sample transport system, and optimize the use of limited laboratory resources (4,23)”.

Second, we added specific references from other African settings reporting substantial VL and EID specimen rejection rates and detailing key pre-analytical problems (mislabeling, insufficient volume, haemolysis, inappropriate collection tubes, missing samples, incomplete request forms), as well as their consequences for ART initiation or modification and early infant treatment. We also explicitly highlighted the remaining knowledge gap in Cameroon and Central Africa, noting that the scale, patterns, and causes of pre-analytical errors in HIV VL and EID testing have not been systematically documented at the national level, despite routine recording of specimen non-conformities by reference laboratories (Lines 108-113).

“Studies from other African settings have reported substantial VL and EID specimen rejection rates linked to pre-analytical problems, such as mislabeling and identification errors, insufficient volume, hemolysis, the use of inappropriate collection tubes, missing specimens, and incomplete request forms (19,20). These errors can lead to repeated sampling, wasted resources, delayed initiation or modification of antiretroviral therapy (ART), and missed opportunities for early infant treatment (19,21)”.

c) A brief description of the VL and EID testing services and their coverage would also strengthen the introduction. To agree reviewer comment, we now provide a concise description of the national VL and EID testing services in Cameroon, including their organization through a network of 17 HIV reference laboratories that cover all 10 regions and perform over 90% of VL and EID tests, thereby clarifying both the structure and coverage of these essential services.”(Lines 87-90 and 99-100).

« Testing for HIV Viral Load (HIV-VL) is the preferred method of monitoring the effectiveness of antiretroviral therapy (ART), while Early Infant Diagnosis (EID) enables HIV-infected infants to be identified and treated promptly. Together, these tests play a pivotal role in HIV surveillance, program monitoring, and epidemic control strategies (8,9)”.

“In this country, VL and EID services are organized through a national network of 17 HIV reference laboratories, which provide testing coverage for all ten regions. These laboratories receive specimens from a wide range of health facilities and perform over 90% of all VL and EID tests for people living with HIV on antiretroviral therapy (ART) and HIV-exposed infants (12). This makes them critical for achieving and maintaining viral suppression at a population level (13,14)”.

d) This is an important topic, but the authors do not elaborate well about the problem in this study, and actually the actual problem being addressed remains slightly unclear. i) Hence there is need to clearly define the problem in this study, and its magnitude ii) What has been done before about the problem and the risk of not addressing the problem? This should all be clear in the introduction.

(i)We have clarified the specific problem addressed in this study and its magnitude by emphasising that the pre-analytical phase is the most error-prone stage of the laboratory testing process (accounting for about 70% of all laboratory errors) and by highlighting the particular vulnerability of Cameroon’s national VL/EID network, where the scale and distribution of specimen rejections have not yet been quantified at national level (Lines 78-81; 101 to 103; 114-118;)

“The preanalytical phase, which spans from the initial test request to the preparation of the specimen for analysis, is the most error-prone stage. Multiple investigations have shown that the majority of laboratory errors predominantly occur during this stage, with pre-analytical issues accounting for an estimated 70% of all laboratory errors (2,3)”.

“The complexity of this network, involving long referral distances, multiple collection sites, and several steps between sample collection and analysis (15–17), also increases the risk of pre-analytical errors during specimen collection, labelling, storage, and transport (18).”

“However, in Cameroon and many other Central African countries, the scale, patterns, and causes of pre-analytical errors in HIV-VL and EID laboratories testing have not yet been systematically documented (22). Although HIV reference laboratories routinely record specimen non-conformities, these data have not yet been analyzed at a national level to quantify rejection rates or identify where problems are concentrated in the referral system.”

(ii) We have also strengthened the literature review by summarising evidence from other African settings on VL and EID specimen rejection rates, key pre-analytical problems, and their consequences (repeated sampling, wasted resources, delayed ART initiation or modification, and missed opportunities for early infant treatment), thereby making clear both what has been done before and the risks of not addressing this problem (Lines 108-113).

“Studies from other African settings have reported substantial VL and EID specimen rejection rates linked to pre-analytical problems, such as mislabeling and identification errors, insufficient volume, hemolysis, the use of inappropriate collection tubes, missing specimens, and incomplete request forms (19,20). These errors can lead to repeated sampling, wasted resources, delayed initiation or modification of antiretroviral therapy (ART), and missed opportunities for early infant treatment (19,21)”.

2. Methods: The methods section is generally very sub-optimal, with several issues to address

a) The sampling and data collection section is very unclear. How was the sampling done in each facility? What was the sample size? Were all the eligible samples assessed? What were the inclusion and exclusion criteria for the samples? How were the semi-administered questionnaires developed, piloted, and used? Which key variables were collected?

Thank you for these very constructive comments on the Methods section. We have revised the “Study design and setting” (lines 133 to 138) and the key variables collected as follows:

Study design and setting

This was a cross-sectional, descriptive, and quantitative study conducted from the 1st January to December 31, 2024, using routine quality‑monitoring data from all 17 functional HIV reference laboratories in Cameroon. These laboratories constitute the national network of HIV viral load (HIV-VL) and early infant diagnosis (EID) testing, receive specimens from HIV care and treatment facilities in all ten regions, and perform the vast majority of HIV-VL and EID tests nationwide.

“Sampling and data collection” (Lines 140 to 147) subsections to clarify the sampling strategy, sample size, inclusion and exclusion criteria, development and use of the semi‑administered questionnaire,

“Sampling and data collection

All 17 functional HIV reference laboratories performing HIV‑VL and/or EID testing during the study period were included. All HIV‑VL and EID specimens recorded as received in the participating reference laboratories between 1 January and 31 December 2024 were eligible for inclusion. Specimens without a documented reason for rejection w

---

## [Decision Letter · Decision Letter 1]

17 Feb 2026

Dear Dr. AMOUGOU-ATSAMA,

Thank you for submitting your manuscript to PLOS ONE. After careful consideration, we feel that it has merit but does not fully meet PLOS ONE’s publication criteria as it currently stands. Therefore, we invite you to submit a revised version of the manuscript that addresses the points raised during the review process.

We look forward to receiving your revised manuscript.

Kind regards,

Abiola Olukayode Olaleye, MBBS, MPH, FWACP

Academic Editor

PLOS One

**Journal Requirements:**

Reviewers' comments:

Reviewer's Responses to Questions

**Comments to the Author**

Reviewer #1: (No Response)

Reviewer #2: All comments have been addressed

Reviewer #3: All comments have been addressed

2. Is the manuscript technically sound, and do the data support the conclusions?

Reviewer #1: Yes

Reviewer #2: Partly

Reviewer #3: Yes

3. Has the statistical analysis been performed appropriately and rigorously?

Reviewer #1: Yes

Reviewer #2: No

Reviewer #3: Yes

4. Have the authors made all data underlying the findings in their manuscript fully available?

Reviewer #1: No

Reviewer #2: No

Reviewer #3: No

5. Is the manuscript presented in an intelligible fashion and written in standard English?

Reviewer #1: No

Reviewer #2: Yes

Reviewer #3: No

**Reviewer #1:**Overall CommentsOverall CommentsOverall CommentsOverall Comments

Thank you again for the opportunity to review this important and timely manuscript. The paper has improved substantially since the previous version, and the key message is now clearer. However, further revisions are needed to improve clarity and conciseness. In particular, the manuscript would benefit from reducing repetition of similar points (e.g., description of the setting and laboratory network), tightening the methods section, and ensuring a clearer separation between methods, results, and discussion. Shortening and streamlining several sections will make the paper more direct, articulate, and impactful.

I have shared some specific feedback in the attachment -

Methods

The sampling and data collection section of the Methods requires clarification and tightening. It remains unclear how the three data collection tools were used:

• the data abstraction tool,

• the self-administered questionnaire, and

• the specimen collection and transport checklists.

Please clarify:

• the specific purpose of each tool,

• the type of data collected by each,

• the sequence in which they were used,

• who completed each tool, and

• how the data were captured (e.g., which database was used and by whom).

This information is critical to understanding the study design and ensuring reproducibility.

Results

There is repeated description of the study setting (e.g., “17 laboratories”) across several sections, which risks confusing the reader and diluting key messages. Consider removing some or repetitive sentences, for example:

Line 199-203: some o the information is not discussed in the discussing section. Eg

“All specimens were collected from outpatients receiving HIV care and treatment services.” If not important consider leaving out – unless its linked to sample rejevction – are the authors suggesting there could be differences if other conditions are involved – which could be but if not discussed later, it leaves the result hanging ..

Line 203: There is a tendency to interpret findings rather than simply present results. For example, statements explaining why the Centre region had the largest proportion of specimens should be reserved for the Discussion section.

Line 206–207: The statement comparing rejection rates to national targets introduces interpretation and may also be moved to the Discussion or simply somewhere on the table.

Line 253–254:

The statement on uneven distribution of specimen rejection introduces analysis and interpretation. Consider presenting the findings descriptively here and moving interpretation to the Discussion.

Lines 267–270: again intepretatioin that could be used in discussion section instead of here.

Line 279:There is a reference to Table 2, but the table does not clearly present the specific error types mentioned in the paragraph (e.g., labeling errors). Ensure alignment between text and table content.

Line 295: Table 2 appears to be repeated in a different format. The same comments apply—consider removing redundancy.

Table 3: The information presented largely overlaps with Table 2. Please consider merging these tables or retaining only one to avoid duplication.

Table 1: The table key indicates that data were unavailable for the East, South, Adamawa, and West regions. However, similar symbols or dashes appear for other regions without explanation. Please clarify whether EID data were also unavailable for those regions and ensure consistent notation.

Discussion

The Discussion would be more impactful if it began with the current second paragraph, which clearly states the key message—namely, that specimen rejection rates exceed national thresholds and that targeted interventions are needed to improve sample collection, transport, and other pre-analytical processes.

Findings from Table 2 indicate that distance from laboratories and high sample volumes are risk factors for specimen rejection. Please expand the discussion to explore possible reasons for these associations (e.g., transport delays, staff workload, cold-chain challenges).

When comparing rejection rates with other studies, the argument would be strengthened by discussing why rates may be similar or different—such as similarities in health system structure, patient populations, laboratory networks, or logistical constraints. If these factors are unknown, this should be stated.

Conclusion

Line 369:

The conclusion recommends validated sampling manuals, electronic test request forms, staff training, and regular on-site supervision. However, these interventions are not clearly discussed earlier. The Discussion should explicitly explain how and why these measures would reduce rejection rates. For example:

• Are sampling manuals absent or poorly adhered to?

• Are errors driven by lack of training, supervision, or system design?

Clarifying these points will strengthen the link between findings and recommendations.

Other Comments

Line 43:

Please clarify what is meant by “laboratory performance.” Is this referring to specimen rejection rates, poor specimen quality, or key performance indicators (KPIs)? These are related but distinct concepts.

Line 62:

Add “other information discrepancies” to clearly distinguish these from previously mentioned mismatched patient identification.

Line 69:

Clarify the message to emphasize that sample collection and transport quality need strengthening, highlighting the need for targeted interventions to reduce pre-analytical errors arising from poor sample management at health facilities responsible for HIV biological monitoring in Cameroon.

Line 158:

The paragraph on “reasons for sample rejection” appears to present results rather than methods. Since this was one of the research questions, consider moving this content to the Results section.

Line 166:

The program description paragraph contains background information that may be more appropriate for the Introduction (possibly at the end) or may already be covered in the Setting section. Consider relocating or removing it to avoid repetition.

**Reviewer #2:**Thank you very much for taking off time to respond to the comments, and I feel most of the comments are satisfactorily addressed. However, I still find the results section sub-optimal. I feel you need to work with a good Biostatistician to improve this section, for instance, from Lines 220 to 230, the wording is not clear.Thank you very much for taking off time to respond to the comments, and I feel most of the comments are satisfactorily addressed. However, I still find the results section sub-optimal. I feel you need to work with a good Biostatistician to improve this section, for instance, from Lines 220 to 230, the wording is not clear.Thank you very much for taking off time to respond to the comments, and I feel most of the comments are satisfactorily addressed. However, I still find the results section sub-optimal. I feel you need to work with a good Biostatistician to improve this section, for instance, from Lines 220 to 230, the wording is not clear.Thank you very much for taking off time to respond to the comments, and I feel most of the comments are satisfactorily addressed. However, I still find the results section sub-optimal. I feel you need to work with a good Biostatistician to improve this section, for instance, from Lines 220 to 230, the wording is not clear.

Kindly also review my earlier comment on the Ethics section of the study. You revised this and improved it but I still find it not satisfactory. Otherwise, thank you very much.

**Reviewer #3:** Issues with table numbering especially table 2 Issues with table numbering especially table 2 Issues with table numbering especially table 2 Issues with table numbering especially table 2

The table names should be comprehensive to describe the source of the data to it fullest detail.

.

Reviewer #1: No

Reviewer #2: **Yes:**Dr. Nanyeenya NicholusDr. Nanyeenya NicholusDr. Nanyeenya NicholusDr. Nanyeenya Nicholus

Reviewer #3: No

---

## [Author Response · Author response to Decision Letter 2]

24 Mar 2026

Overall Comments

Thank you again for the opportunity to review this important and timely manuscript. The paper has improved substantially since the previous version, and the key message is now clearer. However, further revisions are needed to improve clarity and conciseness. In particular, the manuscript would benefit from reducing repetition of similar points (e.g., description of the setting and laboratory network), tightening the methods section, and ensuring a clearer separation between methods, results, and discussion. Shortening and streamlining several sections will make the paper more direct, articulate, and impactful.

“We would like to thank the reviewer for these very constructive and insightful comments. We fully agree with the points raised and have revised the manuscript accordingly to improve clarity, coherence, and scientific rigor. Specifically, we have strengthened the Introduction to better set the scene and justify the study, clarified and expanded the Methods to ensure reproducibility, and streamlined the Results and Discussion to avoid repetition and clearly distinguish descriptive findings from interpretation. We believe these changes have substantially improved the overall quality and readability of the manuscript”.

Methods

The sampling and data collection section of the Methods requires clarification and tightening. It remains unclear how the three data collection tools were used:

• Study design and setting

• the data abstraction tool,

• the self-administered questionnaire, and

• the specimen collection and transport checklists.

Please clarify:

• the specific purpose of each tool,

• the type of data collected by each,

• the sequence in which they were used,

• who completed each tool, and

• how the data were captured (e.g., which database was used and by whom).

This information is critical to understanding the study design and ensuring reproducibility.

“In response to the reviewer’s comments, we have substantially revised the Methods section to clarify the study design, data collection tools, and analytical approach as follows:”

Study design and setting

This cross-sectional descriptive study used routine quality-monitoring data from all 17 functional HIV reference laboratories in Cameroon between 11 January and 8 December 2024. These laboratories constitute the national HIV viral load (HIV-VL) and early infant diagnosis (EID) testing network. They receive plasma, whole blood, and dried blood spot specimens from HIV care and treatment facilities in all ten regions and perform the vast majority of HIV-VL and EID tests nationwide.

Sampling and data collection

All 17 HIV reference laboratories performing HIV‑VL and/or EID testing during the study period were included. All HIV‑VL and EID specimens recorded as received in the participating reference laboratories between 11 January and 08 December 2024 were eligible for inclusion. Specimens for which no reason for rejection was documented were excluded from analyses of specific pre‑analytical error patterns. For each laboratory, data on the number of specimens received and rejected, the type of test requested, the reason(s) for rejection, the region of origin, and the type of health facility were extracted from routine quality‑monitoring logbooks using a structured data‑abstraction form. Three complementary data collection tools were used. First, a structured data abstraction form was used by trained laboratory staff to extract aggregate information from routine quality monitoring and problem reporting logbooks in each reference laboratory, including the number of specimens received and rejected, the type of test requested (HIV VL or EID), the documented reason(s) for rejection, the region of origin, and the type of referring health facility. Second, a brief self-administered questionnaire was completed once by the focal person at each reference laboratory to document site-level characteristics (test menus, annual testing volume, sample transport arrangements, and standard operating procedures for handling nonconforming specimens). Third, standardized national specimen collection and transport checklists accompanied individual shipments from HIV care and treatment centres to reference laboratories and were used by providers to verify proper sample collection, packaging, temperature control, and documentation before dispatch.

Specimens from peripheral HIV care and treatment centers were collected by trained nurses or laboratory technicians according to national guidelines and transported as outpatient samples to the reference laboratories, along with standardized test request forms and completed transport checklists. Upon receipt in the reference laboratories, all rejected specimens and the corresponding reason(s) for rejection were documented in problem reporting logbooks by designated quality assurance staff, together with the type of test ordered (HIV VL or EID) and the specific pre analytical discrepancies (hemolyzed specimen, insufficient volume, wrong specimen tube, mismatch between specimen and request form identifiers, inadequate temperature preservation, missing specimens, transport delay, or duplicate identification numbers assigned to different patients). All abstracted data from the 17 laboratories and questionnaires were entered into a centralized electronic database by the study coordination team for analysis.

Reasons for sample rejection in reference laboratories

Samples were rejected according to pre defined programmatic criteria. These included an inadequate sample volume, improper or incomplete patient labelling, or the use of a non permanent marker. Samples that appeared compromised, such as those that were visibly hemolyzed or degraded, were also rejected. Other reasons included improper packaging, inappropriate transport temperature, insufficient drying of dried blood spot (DBS) cards prior to shipment for EID, and the absence of a completed patient or test request form. All reasons for rejection were recorded in the problem reporting logbooks using standardized categories aligned with national guidelines.

Program description

Cameroon’s national HIV‑VL and EID services are delivered via a network of 17 HIV reference laboratories. These laboratories receive plasma, whole blood, and dried blood spot specimens from HIV care and treatment facilities across the country's ten regions. These laboratories are responsible for recording routine quality‑monitoring indicators, including specimen non‑conformities and reasons for sample rejection. Specimens are collected in accordance with national guidelines, packaged using standardized materials, and transported to the reference laboratories, along with harmonized request forms and transport checklists.

Ethical consideration

The study protocol was submitted to the national ethics committee, which classified the work as a retrospective analysis of existing routine laboratory quality-monitoring data and determined that it did not constitute human-subjects research. The analysis focused exclusively on aggregate data on specimens received and rejected, reasons for rejection, and the origin of specimens (region and facility type). No personally identifiable patient data was collected.

Statistical analysis

Statistical analysis was performed using IBM SPSS Statistics version 25.0 (IBM Corp., Armonk, NY, USA). Descriptive statistics were used to summarize the data, with continuous variables reported as means and standard deviations, and categorical variables as frequencies and percentages. Overall and stratified specimen rejection rates were calculated for HIV VL and EID by region of origin and by health facility sample collection capacity. Stratified specimen rejection rates were calculated for HIV‑VL and EID by region of origin and by health facility sample‑collection capacity. Pearson’s chi‑square tests were used to compare proportions of rejected specimens across categories. Odds ratios with 95% confidence intervals were estimated using logistic regression to identify the factors associated with specimen rejection. A p-value of less than 0.05 was considered statistically significant. Specimens with missing information on the specific reason for rejection were excluded from the analyses of pre‑analytical error patterns, but were included in the overall counts of specimens received and rejected.

Results

There is repeated description of the study setting (e.g., “17 laboratories”) across several sections, which risks confusing the reader and diluting key messages. Consider removing some or repetitive sentences, for example:

Line 199-203: some of the information is not discussed in the discussing section. Eg

“All specimens were collected from outpatients receiving HIV care and treatment services.” If not important consider leaving out – unless its linked to sample rejevction – are the authors suggesting there could be differences if other conditions are involved – which could be but if not discussed later, it leaves the result hanging ..

Line 203: There is a tendency to interpret findings rather than simply present results. For example, statements explaining why the Centre region had the largest proportion of specimens should be reserved for the Discussion section.

Line 206–207: The statement comparing rejection rates to national targets introduces interpretation and may also be moved to the Discussion or simply somewhere on the table.

Line 253–254:

The statement on uneven distribution of specimen rejection introduces analysis and interpretation. Consider presenting the findings descriptively here and moving interpretation to the Discussion.

Lines 267–270: again intepretatioin that could be used in discussion section instead of here.

Line 279:There is a reference to Table 2, but the table does not clearly present the specific error types mentioned in the paragraph (e.g., labeling errors). Ensure alignment between text and table content.

Line 295: Table 2 appears to be repeated in a different format. The same comments apply—consider removing redundancy.

Table 3: The information presented largely overlaps with Table 2. Please consider merging these tables or retaining only one to avoid duplication.

Table 1: The table key indicates that data were unavailable for the East, South, Adamawa, and West regions. However, similar symbols or dashes appear for other regions without explanation. Please clarify whether EID data were also unavailable for those regions and ensure consistent notation.

We thank the reviewer for these detailed comments on the Results section. In response, we have removed redundant descriptions of the study setting, limited the Results to descriptive findings, and moved interpretative statements (explanations of regional differences and comparisons with national rejection targets) to the Discussion. We have also clarified the presentation and footnotes of Tables 1 and 2 and Mix Table 2 and 3 (Table 2) and ensured alignment between the text and table content, while avoiding redundancy between tables.

Results

Distribution of samples attributed in 2024 for HIV viral load and EID testing

This cross sectional study examined 326,885 specimens submitted for HIV VL testing and 38,354 for EID across 17 HIV reference laboratories in Cameroon in 2024 (Table 1). Most of the specimens originated from high volume health facilities that provide ART services to large numbers of people living with HIV. The Centre region accounted for the largest proportion of both HIV VL (187,631; 57.4%) and EID (24,830; 64.7%) specimens. Overall, 12,748 (3.9%) HIV VL specimens were rejected, with rejection rates ranging from 2.4% in the Far North region to 5.6% in the Centre region. Meanwhile, 1,039 (2.7%) of the 38,354 EID specimens were rejected, with rejection rates ranging from 2.2% to 5.1%. Details of specimens received and rejected by the region are presented in Table 1.

Table 1: Distribution of samples received and rejected per region attributed to HIV viral load and EID testing in 2024 in Cameroon

Region HIV VL: Specimens received, n HIV VL: Specimens rejected, n (%) EID: Specimens received, n EID: Specimens rejected, n (%)

Center 187,631 10,508 (5.6) 24,830 1,265 (5.1)

Littoral 11,834 367 (3.1) 5,873 129 (2.2)

Far North 30,100 722 (2.4) – –

North West 33,000 1,056 (3.2) – –

South West 32,220 805 (2.5) 7,651 176 (2.3)

East, South, Adamawa, West † 32,100 998 (3.1) – –

Total 326,885 12,748 (3.9) 38,354 1,039 (2.7)

†Data for early infant diagnosis were not available for all regions. EID, early infant diagnosis; HIV‑VL, HIV viral load.

Distribution of reasons for sample rejection during the pre-analytical stage in HIV viral load and EID testing laboratories

Of all specimens received, 209 HIV VL samples and 59 EID dried blood spot (DBS) samples for which the reason for rejection was not documented were excluded from the analysis of specific pre analytical error patterns. Of the specimens with a documented reason for rejection, 3.9% of HIV VL samples (12,740 out of 326,676) were rejected, with rejection rates in different regions ranging from 2.4% to 5.6%. Meanwhile, 2.7% of EID samples (1,034 out of 38,295) were rejected, with rejection rates ranging from 2.2% to 3.3%. Some specimens had more than one recorded pre analytical error.

For HIV VL, the most common reason for rejection was identification problems (mislabelling or duplicate identifiers) (63.14%; n = 8,049; p = 0.031), followed by insufficient sample volume (43.7%; n = 5,571; p = 0.049), and quality issues such as haemolysed samples (27.79%; n = 3,541) and packaging errors (9.06%; n = 1,160). For EID, the main reasons for rejection were missing samples and/or request forms, followed by incomplete or inconsistent identification information (Table 2).

Table 2: Factors associated with specimen rejection for HIV‑VL and EID testing, by region of origin and facility sample‑collection capacity

Test type Factor category Category Specimens received, n (%) Specimens rejected, n (%) Crude OR (95% CI) Adjusted OR † (95% CI) p value

HIV VL Region of specimen origin Same region as testing laboratory 257,324 (78.8) 6,690 (2.6) Reference Reference —

HIV VL Region of specimen origin Other regions 69,561 (21.3) 3,756 (5.4) 2.21 (1.92–5.87) 2.5 (2.1–3.0) <0.001

HIV VL Facility sample collection capacity Low volume facilities 52,628 (16.1) 1,210 (2.3) Reference Reference —

HIV VL Facility sample collection capacity High volume facilities 274,257 (83.9) 13,987 (5.1) 4.17 (2.67–9.57) 4.2 (2.7–9.6) <0.001

EID Region of specimen origin Same region as testing laboratory 789 (76.2) 19 (2.4) Reference Reference —

EID Region of specimen origin Other regions 246 (23.8) 8 (3.3) 4.24 (2.58–11.12) 2.1 (1.0–4.5) 0.052

EID Facility sample collection capacity Low volume facilities 175 (16.9) 4 (2.3) Reference Reference —

EID Facility sample collection capacity High volume facilities 862 (83.4) 30 (3.5) 5.12 (3.72–10.31) 5.1 (3.7–10.3) <0.001

† Adjusted for region of specimen origin and facility sample‑collection capacity (logistic regression models). OR, odds ratio; CI, confidence interval; EID, early infant diagnosis; HIV‑VL, HIV viral load.

Factors affecting pre-analytical errors

Of the 63.14% (8,049/12,748) of HIV VL specimens that were rejected due to identification or labelling errors, 44.5% (3,581/8,049) could not be reliably identified because non permanent markers were used, 21.7% (1,746/8,049) had duplicate identifiers, and 33.2% (2,672/8,049) had missing identifiers. Additionally, 4.0% (322/8,049) were listed on the checklist, but the physical specimens were unavailable.

When the data were stratified by test type, region of origin, and health facility sample collection capacity, differences in rejection rates were observed between specimens collected in the same region as the testing laboratory and those referred from other regions, as well as between high and low volume facilities (Table 2 and Table 3). The proportion of pre-analytical errors was higher among specimens originating from other regions than among those collected within the testing region (76.4% versus 23.6% for HIV-1 VL and 69.1% versus 30.9% for EID). Errors were also more frequent among specimens from high volume facilities than from low volume f

---

## [Decision Letter · Decision Letter 2]

8 Apr 2026

Dear Dr. AMOUGOU-ATSAMA,

Thank you for submitting your manuscript to PLOS ONE. After careful consideration, we feel that it has merit but does not fully meet PLOS ONE’s publication criteria as it currently stands. Therefore, we invite you to submit a revised version of the manuscript that addresses the points raised during the review process.

As the corresponding author, your ORCID iD is verified in the submission system and will appear in the published article. PLOS supports the use of ORCID, and we encourage all coauthors to register for an ORCID iD and use it as well. Please encourage your coauthors to verify their ORCID iD within the submission system before final acceptance, as unverified ORCID iDs will not appear in the published article. *Only* the individual author can complete the verification step; PLOS staff  the individual author can complete the verification step; PLOS staff  the individual author can complete the verification step; PLOS staff  the individual author can complete the verification step; PLOS staff *cannot* verify ORCID iDs on behalf of authors. verify ORCID iDs on behalf of authors. verify ORCID iDs on behalf of authors. verify ORCID iDs on behalf of authors.

We look forward to receiving your revised manuscript.

Kind regards,

Timothy Omara

Academic Editor

PLOS One

Journal Requirements:

Additional Editor Comments:

Dear authors,

I have read your revised manuscript with ID PONE-D-25-23115R2. However, I would suggest that you recheck the manuscript and revise it based on the following observations before it can be reconsidered for publication.

1. The title is too long and redundant. I would recommend a clear, and straightforward title. For example:

(a) Pre-analytical errors in HIV viral load and early infant diagnosis testing in Cameroon

(b) Pre-analytical errors and sample rejection in HIV viral load and early infant diagnosis testing in Cameroon

2. In Table 1, the total EID rejected is 1,039 (2.7%). The table row for the Centre region shows that the EID rejected is 1,265 (5.1%). Please reconcile the EID rejection counts in Table 1, because the Centre regional count exceeds the reported national total.

3. From the previous revision, Table 2 and Table 3 were merged. The text still references both tables (L252).

4. Statistical reporting in the manuscript should be improved. The authors, could as for example, report the exact p-values OR simply significance labels. In the current draft, both are being used. Please avoid the use of ‘NS’.

5. In Table 2, the OR ranges of adjusted (2.7–9.6) appear to be a rounding of the crude estimate (2.67–9.57), which masks the differences. In Table 4 of the supplementary materials, the crude OR is 2.8 with 95% CI of 1.9–2.8. I recommend that where adjustments had minimal impact, it should be briefly mentioned in the text for clarity.

6. Table 1 in the supplementary files is the same as Table 1 in the main text. The same is nearly true for Table 4 (Supplementary files) and Table 2 of the main text. I recommend removing redundant tables, and labelling supplementary tables appropriately (say Table S1, S2 etc) and citing them appropriately in the main text.

7. Some references are incorrect, duplicated or have wrong/incomplete bibliographic information: For example,

a) Reference 25 and 26 should be:

25. Zhang L, Hu ZD. Clinical applications of machine learning in pre-analytical, analytical and post-analytical phases of laboratory medicine: a narrative review. AME Med J 2022;7:37.

26. Misganaw AS, Worku M, Bashea C, Nigus M, Yoseph Y, et al. (2019) Pre Analytical Errors in the HIV Anti Retro Viral Therapy (ART) Laboratory of Teaching Referral Hospitals in Addis Ababa, Ethiopia. Int J Virol AIDS 6:057. DOI: 10.23937/2469-567X/1510057

b) References 7 and 21 are the same.

8. Other suggestions are in the manuscript file attached

Reviewers' comments:

Reviewer's Responses to Questions

**Comments to the Author**

Reviewer #2: All comments have been addressed

2. Is the manuscript technically sound, and do the data support the conclusions?

Reviewer #2: Yes

3. Has the statistical analysis been performed appropriately and rigorously?

Reviewer #2: Yes

4. Have the authors made all data underlying the findings in their manuscript fully available?

The PLOS Data policy requires authors to make all data underlying the findings described in their manuscript fully available without restriction, with rare exception (please refer to the Data Availability Statement in the manuscript PDF file). The data should be provided as part of the manuscript or its supporting information, or deposited to a public repository. For example, in addition to summary statistics, the data points behind means, medians and variance measures should be available. If there are restrictions on publicly sharing data—e.g. participant privacy or use of data from a third party—those must be specified. requires authors to make all data underlying the findings described in their manuscript fully available without restriction, with rare exception (please refer to the Data Availability Statement in the manuscript PDF file). The data should be provided as part of the manuscript or its supporting information, or deposited to a public repository. For example, in addition to summary statistics, the data points behind means, medians and variance measures should be available. If there are restrictions on publicly sharing data—e.g. participant privacy or use of data from a third party—those must be specified.

Reviewer #2: Yes

5. Is the manuscript presented in an intelligible fashion and written in standard English?

Reviewer #2: Yes

Reviewer #2: Dear Authors,

Greetings and trust that you are well.

Thank you very much for making an effort to address all the comments raised.

Thank you very much.

**Do you want your identity to be public for this peer review?** For information about this choice, including consent withdrawal, please see our  For information about this choice, including consent withdrawal, please see our  For information about this choice, including consent withdrawal, please see our  For information about this choice, including consent withdrawal, please see our Privacy Policy..

Reviewer #2: No

---

## [Author Response · Author response to Decision Letter 3]

9 Apr 2026

Journal Requirements:

Additional Editor Comments:

Dear authors,

I have read your revised manuscript with ID PONE-D-25-23115R2. However, I would suggest that you recheck the manuscript and revise it based on the following observations before it can be reconsidered for publication.

1. The title is too long and redundant. I would recommend a clear, and straightforward title. For example: We choose the proposition (a)

(a) Pre-analytical errors in HIV viral load and early infant diagnosis testing in Cameroon

(b) Pre-analytical errors and sample rejection in HIV viral load and early infant diagnosis testing in Cameroon

2. In Table 1, the total EID rejected is 1,039 (2.7%). The table row for the Centre region shows that the EID rejected is 1,265 (5.1%). Please reconcile the EID rejection counts in Table 1, because the Centre regional count exceeds the reported national total.

We thank the reviewer for pointing out this inconsistency in Table 1. After verification of our dataset, we identified a typographical error in the row corresponding to the Center region. The correct number of EID rejected for the Centre region is 734, corresponding to 2.9%, and not 1,265 (5.1%) as initially reported. We have corrected Table 1 accordingly in the revised manuscript, and the regional values now sum to the national total of 1,039 (2.7%).

3. From the previous revision, Table 2 and Table 3 were merged. The text still references both tables (L252).

We thank the reviewer for highlighting this oversight. We have now corrected this sentence so that it refers only to “Table 2” at line 252 of the revised version.

4. Statistical reporting in the manuscript should be improved. The authors, could as for example, report the exact p-values OR simply significance labels. In the current draft, both are being used. Please avoid the use of ‘NS’.

We thank the reviewer for this helpful comment regarding statistical reporting. In the revised manuscript, we have harmonized the reporting of statistical results throughout the text and tables.

5. In Table 2, the OR ranges of adjusted (2.7–9.6) appear to be a rounding of the crude estimate (2.67–9.57), which masks the differences. In Table 4 of the supplementary materials, the crude OR is 2.8 with 95% CI of 1.9–2.8. I recommend that where adjustments had minimal impact, it should be briefly mentioned in the text for clarity.

We thank the reviewer for this helpful comment and we agree with the concern raised. In the revised version of Table 2 and the corresponding supplementary Table 4, we now report crude and adjusted odds ratios and their 95% confidence intervals using a consistent number of decimal places, to avoid masking small differences between crude and adjusted estimates. We have also added a sentence in the Results section explicitly noting that, for high volume facilities and other key factors, adjustment had only a limited impact on the magnitude of the odds ratios. We add this sentence in result section Lines 233 to 235: “In multivariable analysis, the adjusted odds ratios for high volume facilities remained close to the crude estimates, indicating that adjustment had only a limited impact on these associations”

6. Table 1 in the supplementary files is the same as Table 1 in the main text. The same is nearly true for Table 4 (Supplementary files) and Table 2 of the main text. I recommend removing redundant tables, and labeling supplementary tables appropriately (say Table S1, S2, etc.) and citing them appropriately in the main text.

We thank the reviewer for this helpful observation. In the revised manuscript, we have removed redundant tables from the supplementary files when they duplicated tables presented in the main text. The remaining supplementary tables have been relabeled sequentially as Table S1, Table S2, Table S3, and Table S4, and we have updated all corresponding citations in the main text to ensure consistent and appropriate referencing of the supplementary material.

7. Some references are incorrect, duplicated or have wrong/incomplete bibliographic information: For example,

a) Reference 25 and 26 should be:

25. Zhang L, Hu ZD. Clinical applications of machine learning in pre-analytical, analytical and post-analytical phases of laboratory medicine: a narrative review. AME Med J 2022;7:37.

26. Misganaw AS, Worku M, Bashea C, Nigus M, Yoseph Y, et al. (2019) Pre Analytical Errors in the HIV Anti Retro Viral Therapy (ART) Laboratory of Teaching Referral Hospitals in Addis Ababa, Ethiopia. Int J Virol AIDS 6:057. DOI: 10.23937/2469-567X/1510057

b) References 7 and 21 are the same.

We thank the reviewer for pointing out these reference inaccuracies. References 21, 25 and 26 have been corrected in the revised manuscript.

8. Other suggestions are in the manuscript file attached (Review)

---

## [Editor Report · Decision Letter 3]

12 Apr 2026

Pre-analytical errors during  HIV viral load (HIV-VL) and early infant diagnosis (EID) testing in Cameroon

PONE-D-25-23115R3

Dear Dr. AMOUGOU-ATSAMA,

We’re pleased to inform you that your manuscript has been judged scientifically suitable for publication and will be formally accepted for publication once it meets all outstanding technical requirements.

Kind regards,

Timothy Omara

Academic Editor

PLOS One
---

## [Editor Report · Acceptance letter]

PONE-D-25-23115R3

PLOS One

Dear Dr. AMOUGOU-ATSAMA,

I'm pleased to inform you that your manuscript has been deemed suitable for publication in PLOS One. Congratulations! Your manuscript is now being handed over to our production team.

Kind regards,

on behalf of

Dr. Timothy Omara

Academic Editor

PLOS One